# The Body as a Battleground: A Qualitative Study of the Impact of Violence, Body Shaming, and Self-Harm in Adolescents with a History of Suicide Attempts

**DOI:** 10.3390/ijerph22060859

**Published:** 2025-05-30

**Authors:** Marianne Rizk-Hildbrand, Tara Semple, Martina Preisig, Isabelle Haeberling, Lukasz Smigielski, Dagmar Pauli, Susanne Walitza, Birgit Kleim, Gregor E. Berger

**Affiliations:** 1Department of Child and Adolescent Psychiatry and Psychotherapy, University Hospital of Psychiatry, University of Zurich, 8032 Zurich, Switzerland; 2Neuroscience Center Zurich, University of Zurich and the ETH Zurich, 8057 Zurich, Switzerland; 3Zurich Center for Integrative Human Physiology, University of Zurich, 8032 Zurich, Switzerland; 4Department of Adult Psychiatry and Psychotherapy, Psychotherapy and Psychosomatics, Psychiatric University Hospital, University of Zurich, 8032 Zurich, Switzerland; 5Experimental Psychopathology and Psychotherapy, Department of Psychology, University of Zurich, 8050 Zurich, Switzerland

**Keywords:** suicide attempt, narrative interview, qualitative research, body dissatisfaction, body shaming, violence, self-harm, acute risk

## Abstract

Perceived experiences of violence, trauma, body dissatisfaction, and body shaming emerge as interconnected factors contributing to suicide attempts among adolescents. There is a critical need to improve the understanding and prediction of suicide attempts in this vulnerable population. In this study, a primarily qualitative design was employed, supported by descriptive quantitative elements, such as code frequencies and word clouds, to explore in-depth narrative interviews of adolescents who attempted suicide. Thematic content analysis was utilized to analyze the transcripts of these narrative interviews. The study sample consisted of 22 adolescents (M_age_ = 16.98 years; SD = 1.624; 77% males, 19% females, 4% non-binary or transgender). The content analysis revealed two significant body-related risk factors for suicide attempts: experiences of violence or trauma and body dissatisfaction, which were associated with maladaptive coping strategies, including self-harm, substance use, and eating disorders. Acute somatic warning signs such as dissociative states, loss of control, and disconnection from one’s body suggested pronounced psychophysiological dysregulation. The findings underscore the body as a battleground, where emotional pain related to bodily experiences and perceptions manifests significantly. Future research should integrate subjective body-related experiences in risk assessments and interventions targeting adolescent suicidal behaviors.

## 1. Introduction

In recent years, suicidal behavior and non-suicidal self-injury have become significant public health issues for individuals under 25. According to the World Health Organization [1,2], suicide is the third leading cause of death worldwide among those aged 15–29 years. A UNICEF study conducted in Switzerland and Liechtenstein reported that 8.3% of the 1197 adolescents with a mean age of 16.5 years surveyed had attempted suicide at least once [3]. Comparable rates were found in a large-scale American study of 17,232 adolescents aged 12 to 17 years which reported that, in 2021, 13.3% of females and 6.6% of males had attempted suicide, and 1952 suicide deaths were recorded among adolescents aged 14 to 18 years [4].

Adolescence is a period marked by substantial physical, emotional, and social transformations, and, therefore, represents a time of heightened vulnerability [5]. These transformations can lead to various challenges, which can manifest as feelings of inadequacy, confusion, defeat, or entrapment, often culminating in depression, hopelessness, or suicidal ideation [6,7]. Several studies have reported a significant positive correlation between perceived stress and suicidal behavior in adolescence. In this context, perceived social support, higher hope and optimism, and better problem-solving skills have been identified as moderating protective factors [8,9]. At the same time, particularly adolescents with poor emotion regulation are at a higher risk of suicidal behavior due to stress [10]. Accordingly, the development and practice of emotion regulation strategies, such as acceptance, problem-solving, and cognitive reappraisal, have been shown to function as adaptive strategies [11,12]. Furthermore, the use of mindfulness-based strategies, such as breathing practices, meditation, progressive muscle relaxation, and mindfulness, can be promising to reduce stress, anxiety, and depressive symptoms [13,14]. Finally, supportive social interactions, activities, and higher self-esteem have been shown to mitigate some of the harmful effects of stress [15]. Research on mindfulness in adolescents highlights its potential as a protective factor that fosters, on the one hand, life satisfaction, resilience, and positive self-esteem [16], and on the other hand, adaptive eating habits, body appreciation, and body satisfaction-related coping strategies [17]. Furthermore, adolescent changes and stressors can also influence physical functions. Elevated cortisol levels as a result of stress can trigger a cycle of various detrimental physiological processes, including the impairment of cognitive functions [18] and damage to the nervous and digestive systems, which can lead to disturbed sleep (insomnia) and altered appetite or eating disorders [19].

The impact of body-related stressors in adolescence is further intensified by the rising incidence of cyberbullying victimization, which further undermines adolescents’ mental well being. Existing research has highlighted the growing dual influence of digital media on adolescents’ self-image and mental health. Negative experiences include exposure to negative content, cyberbullying, constant social comparison, fear, and addictive use patterns [20]. A mediation analysis conducted by Feng et al. [21] investigated the associations between screen time, cyberbullying victimization, and suicidal behaviors among 13,982 American high school students aged 15 to 17. The findings revealed that exposure to cyberbullying was significantly associated with increased rates of suicidal ideation and suicide attempts among adolescents. Furthermore, research increasingly suggests that adolescents are particularly sensitive to external evaluations, including those concerning body image. They are often exposed to unrealistic body standards on social media, which can reinforce harmful behaviors, increase body dissatisfaction, and encourage disordered eating [22]. Among adolescents, body dissatisfaction and body shaming—whether in the form of peer criticism, bullying, social media pressures, or internalized societal ideals—have been linked globally to elevated risks of mental health issues, such as depression, anxiety, and suicidal ideation [23,24,25]. In this digital environment, the body becomes not only a target of self-scrutiny but also a medium of public performance, heightening psychological vulnerability among already at-risk adolescents.

While the effects of body shaming have not been deeply researched, there is evidence that the resulting body dissatisfaction is associated with self-hatred and eating psychopathology [26], particularly in adolescents with low body appreciation and an internalized thin beauty ideal [27]. The findings of the systematic review by Day et al. [28] highlight that adolescents who experience body shaming and bullying are at a heightened risk of developing eating disorders or body dissatisfaction. Furthermore, suicidal ideation occurs in 51% of adolescent individuals with eating disorders, and suicide attempts occur in 21% of these individuals [29]. Consequently, body dissatisfaction is not a mere complaint but a significant stressor that fosters a “battleground” within adolescents, where intense conflicts arise concerning self-image, self-worth, and physical appearance.

The existing literature indicates that trauma, specifically childhood sexual abuse, can trigger self-injurious behaviors [30]. This highlights the need to consider the body’s experiences in the context of self-harm and suicidal ideation. These findings are consistent with established theoretical frameworks that emphasize the embodied nature of psychological trauma. In “*The Body Keeps the Score*” [31], van der Kolk describes how traumatic experiences are stored in the body and can impact physical and emotional well being long after the events themselves. Similarly, Maté [32] underscores in his book “*When the Body Says No*” the physiological imprint of chronic stress and emotional repression on physical health. These perspectives support the view that the body should be understood not only as a site of vulnerability but also as a medium through which psychological suffering is processed and expressed.

Additionally, a multitude of research findings indicate that particularly exposure to violence is a risk factor for suicidal behavior among adolescents [33,34]. Wang et al. [35] examined the interplay between violence and suicidal ideation among depressive adolescents and concluded that particularly recent violence may provide critical information for assessing suicide risk. Nevertheless, the underlying mechanisms are insufficiently understood and require further research.

## 2. Purpose

This study aims to explore adolescents’ relationships with their bodies and the relevance they attribute to their physical selves within the context of their suicide attempts. A primary focus is on understanding body-related experiences, such as body shaming and violence, from the adolescents’ perspective. What distinguishes this research is its exploration of how adolescents articulate body-related risk factors and contextualize them within their narratives of suicide attempts. By analyzing these narratives, this study seeks to uncover the significance adolescents assign to their bodies and how they navigate negative body-related experiences, which is crucial given the complex interplay between body image, mental health, and suicidal behavior.

Despite the existence of numerous models and theories to explain suicidal ideation and behavior, such as the Interpersonal Theory of Suicide by Joiner [36,37] or the Diathesis-Stress Model by Zuckerman [38], the accurate prediction of suicide attempts remains limited. A key challenge is the unpredictability of suicide timing [39], as 30% of completed suicides among adolescents occurred with no identifiable risk factors [40].

This gap highlights the urgent need for novel approaches that move beyond established risk factor analysis and instead gain a deeper understanding of the adolescents’ own perspectives on their triggers for suicidal behavior. An inductive approach appears to be an appropriate method, as it allows an analysis of the adolescents’ perspective without being biased by a predefined research question.

Additionally, analyzing adolescents’ subjective perceptions of their bodies aims to provide a deeper understanding of how dysfunctional body-related stress responses—such as drug use, eating disorders, or self-harm—may contribute to the progression from suicidal ideation to suicide attempts. Existing qualitative research on adolescent suicidal ideation and behavior, summarized in the systematic review by Grimmond et al. [41], has primarily focused on emotional, cultural, and interpersonal stressors. In contrast, the role of the body in suicidal ideation and behavior has been largely overlooked, whereas significant research exists that thoroughly examines suicidal ideation in the context of gender dysphoria [42]. Evidence suggests that body-related factors, such as gender dysphoria [43], non-standard body weight [44], eating disorders [45], and potentially body-related factors, such as bullying [46], alcohol consumption, and sleep disturbances [47], may play a significant role. However, to the best of our knowledge, to date, no studies have examined the body comprehensively as a central contributor to suicidal ideation, considering that body-related stressors such as violence, body shaming, and body dissatisfaction may lead to suicidal behavior.

This study addresses this gap by qualitatively exploring adolescents’ subjective perceptions of body-related stress and its role in suicidality and quantitatively evaluating the frequencies of reported body-related risk factors. Through narrative interviews with adolescents who have recently attempted suicide, this research aims to identify body-related risk factors and warning signs, offering insight into how negative body-related experiences contribute to psychological distress.

While this study is primarily qualitative, quantitative descriptive data, such as code frequencies and linguistic analysis (e.g., word clouds), were integrated as a form of methodological triangulation to support the interpretation of the narratives. These quantitative descriptive data include sociodemographic variables (e.g., age, gender, psychiatric diagnoses) and frequency analyses of narrative codes. These were used to identify recurring patterns and to triangulate qualitative insights with structured data. This dual-methodological approach is envisioned not only to enhance our predictive accuracy regarding adolescent suicidal actions but also to inform the development of targeted interventions. By centering on adolescents’ perceptions and coping strategies, this research aims to transcend traditional risk factor analyses, ultimately contributing to patient-centered approaches that foster positive body image and resilience among adolescents. Such interventions are vital for equipping adolescents with the necessary tools to manage body-related stressors and promote healthier self-perceptions.

In conclusion, this study aims to shed light on the critical interplay between body image and suicide risk in adolescents, advocating for the need to elevate adolescents’ voices in the dialogue surrounding their mental health and well being. By doing so, it aims to pave the way for future therapeutic strategies that prioritize self-acceptance and body positivity in mental health interventions for this vulnerable population.

## 3. Materials and Methods

### 3.1. Setting and Data Source

The data for the present study were collected through narrative interviews as part of routine clinical practice within the Adolescent Attempted Suicide Short Intervention Program, AdoASSIP [48]. This program is a specialized short-term therapy for adolescents following a suicide attempt, integrated into the emergency services of the Department of Child and Adolescent Psychiatry and Psychotherapy at the Psychiatric University Hospital of Zurich. The primary goal of the AdoASSIP is to reduce the risk of reattempted suicides by raising the understanding of the “suicidal mode”. The suicidal mode, as articulated by Rudd [49], is a cognitive–emotional state characterized by a narrowed focus on suicidal thoughts and a suicidal belief system marked by pervasive hopelessness. Within the ASSIP framework—the adult counterpart of the AdoASSIP—this concept of the suicidal mode is further elaborated to describe a psychological state, where individuals often feel trapped and perceive suicide as the only escape from overwhelming emotional pain [50]. Both the ASSIP and the AdoASSIP are based on the assumption that, once the suicidal mode has been activated, it can easily be reactivated in future crises. Consequently, specific therapeutic intervention is essential to disrupt this pattern [48,50].

The AdoASSIP consists of four therapy sessions designed to craft a personalized case concept that includes the patient’s needs, warning signs, goals, and an emergency plan. The first AdoASSIP session contains a video-recorded narrative interview where patients recount the story of their suicide attempt from their own perspective. The therapist initiates the narrative by asking, “*Tell me the story of your suicide attempt in your own words*”. If necessary, open-ended questions may be used to prompt the patient’s further elaboration (e.g., “*Can you tell me more? Can you describe this day in slow motion*?”).

Narrative interviewing, as employed in this study, is a qualitative research method that aims to gather detailed accounts of individuals’ experiences through storytelling [51]. In the context of clinical and therapeutic interventions, narrative interviews serve as a valuable method for identifying the circumstances surrounding suicidal behavior. Research indicates that narrative interviews are particularly effective in eliciting deep insights into complex issues, thereby enhancing understanding of the self, lived experiences, and constructed interpretations [52]. The literature supports the effectiveness of employing narrative interviews as an integral component of qualitative research, with numerous studies highlighting their role in fostering a deeper understanding of personal experiences and enhancing the richness of narrative data [53,54]. Consequently, in our study, a narrative approach was adopted to analyze the subjective challenges faced by individuals who have attempted suicide. By integrating narrative techniques, the AdoASSIP not only assists in identifying potential warning signs but also facilitates the recognition of distinct suicidal patterns that are critical for understanding the patient’s experience.

### 3.2. Participants

The data used in the analysis were derived from adolescent patients enrolled in the AdoASSIP who completed at least the first session of the therapy program. Inclusion criteria for participation in the program included ages 12–17, recent suicide attempts, informed consent of both the adolescent and a legal guardian, sufficient German language skills, and sufficient clinical stability (no acute suicide risk). Additional consent for the research use of the video recordings was obtained voluntarily. This study was approved by the Cantonal Ethics Committee Zurich (BASEC-No. 2024-00724).

Between January 2020 and December 2024, a total of 96 narrative interviews were conducted, with 89 adolescents completing the program. Of these, 69 provided consent for the research use of their video recordings. For the analysis, the first 22 interviews that fulfilled the requirements of the AdoASSIP manual [48] were selected. A critical aspect of the selection criteria was the strict adherence by therapists to the principles inherent in narrative interviewing methodologies. Specifically, therapists employed open-ended, story-generating questions, ensuring minimal directional influence on the participants’ narratives [55]. Furthermore, interviews with high audiovisual quality were prioritized to provide an accurate transcription.

The final selection comprised 22 video-recorded interviews, yielding a demographic representation of 77% female, 19% male, and 4% identifying as other. This distribution aligns closely with the overall demographic profile of all AdoASSIP participants in Zurich, which consisted of 68% female, 23% male, and 9% classified as other. The average age of the sample was 16.98 years (SD = 1.62), and participants ranged in age from 13 to 17 years. Participants had been previously diagnosed with at least one mental disorder, with 81.82% exhibiting at least one comorbid disorder. These diagnoses were categorized as Axis I diagnoses in the sense of major mental disorders, following ICD-10 diagnostic codes [56]. A maximum of four diagnoses were assigned per patient. Among those with a single diagnosis, all were affected by depression or post-traumatic stress disorder (PTSD). Depressive disorders were the most prevalent (63.64%), followed by borderline personality disorder (27.27%), anxiety disorders (27%), ADHD (22%), PTSD (18%), harmful substance use (18%), obsessive–compulsive disorder (9%), eating disorders (9%), and bipolar disorder (9%). Regarding suicide attempts, 31.82% of the sample had attempted suicide once, 31.82% had attempted two or three times, and 36.36% had attempted suicide four or more times.

According to assessments by AdoASSIP therapists, 20 of 22 participants were in suicidal mode at the time of their attempt [49]. In contrast, in only 2 out of 22 participants, no suicidal mode could be identified; these individuals also described their suicide attempt as a deliberate and conscious decision rather than an impulsive act driven by an altered mental state. Most participants either attempted suicide only once (7 of 22) or made four or more attempts (8 of 22). Nearly all participants (20 out of 22) engaged in some level of planning prior to the attempt. Additionally, almost half (9 of 22) wrote farewell letters in advance, and 3 of 22 announced their intent to die by suicide beforehand. The most frequently chosen suicide method was intoxication (16 of 22), followed by attempted jumps from heights (6 of 22). Approximately two-thirds of participants (15 of 22) concealed their suicide attempts, and 9 of 22 required medical treatment afterward. Some individuals (6 of 22) had to be resuscitated or treated in an intensive care unit.

### 3.3. Procedure

The video-recorded narratives were transcribed by graduate students in psychology following the GAT-2 transcription system [57] and the derived transcription conventions [58], which ensure precise and systematic documentation of spoken and nonverbal language. These guidelines emphasize the importance of linguistic accuracy, the recording of pauses, and the notation of significant nonverbal cues, such as laughter and crying.

The video recordings of the narrative interviews ranged from 29 to 64 min in duration, with transcripts varying in length from 2110 to 11,366 words (M = 4826 words, SD = 2270), excluding the closing suicide risk management section, which was not transcribed. Each transcript was coded with segment counts ranging from 69 to 304 (M = 165 segments, SD = 62.8). Since most of the interviews were conducted in Swiss German, which is not written in its standard form, the narratives were first transcribed into High German with the highest possible accuracy. The anonymized transcripts were subsequently imported into MAXQDA software [59] for systematic analysis, supplemented by essential demographic data obtained from clinical records, including gender, age, and clinical diagnosis. Quantitative outputs, such as code frequencies and word clouds, were generated within MAXQDA [59] to complement the qualitative analysis. These served a descriptive and supportive role rather than being subject to inferential statistical testing. Qualitative content analysis was applied to gain deeper insights into underlying themes and contextual meanings. Thematic Coding, based on the approach articulated by Glaser and Strauss [60], was used to systematically analyze the data. This inductive approach is particularly well suited for identifying recurring patterns and relationships without being constrained by pre-existing hypotheses or categories. The overarching goal was to achieve a comprehensive understanding of the risk factors and warning signs related to adolescent suicidal behavior by systematically organizing codes into coherent categories.

The thematic content analysis began with the first author and a scientific co-worker independently familiarizing themselves with the data by inductively coding the first five narratives. This initial phase focused on identifying spontaneously reported and subjectively experienced difficulties and stressors leading to suicide attempts. Subsequently, generated codes were grouped into overarching categories. A collaborative exchange of the two coders followed, during which codebooks were compared, discrepancies were identified and resolved, a consensus codebook was created, and a rule-based coding procedure was established to ensure consistency. Next, both coders independently analyzed three additional narratives, reducing discrepancies and making minor adjustments to the codebook. The initial eight narratives were then re-coded using the finalized consensus codebook. This iterative process resulted in developing a consensus codebook, which was subsequently refined through frequent discussions within the research team.

This process ultimately culminated in formulating the primary research topic: “The Body as a Battleground”. Additional themes, such as therapy experiences or the perceived contagion of suicidal ideation and behavior, were identified and coded. The following phases of coding focused on body-related stressors. Finally, the first author re-examined all transcripts, applying the consensus codebook and creating new body-related codes as they appeared relevant. After completing the coding process, the hierarchical code system was restructured into a final codebook (see Table 1), with some codes being renamed or merged to enhance coherence and analytical clarity. The resulting body-related codebook is structured into ten supercodes and 30 subcodes, as illustrated in Table 1.

Since the transcriptions and coding processes were conducted in German, the codebook and selected quotations were finally translated into English using advanced translation tools [61] and AI-based platforms [62,63]. Particular attention was paid to preserving the original text’s core meaning and contextual nuances rather than strictly adhering to word-for-word translation. This approach aimed to maintain the integrity and accuracy of the participants’ narratives while ensuring cross-linguistic comprehension.

The codebooks used for data analysis in this study can be made available upon request. Given the vulnerability of the adolescent population and the ethical responsibility to protect their privacy, complete anonymization of the transcripts would not be sufficient to ensure confidentiality. Therefore, the transcripts of the narrative interviews cannot be shared due to the content’s highly sensitive and personal nature.

## 4. Results

In addition to the qualitative dimension, this study incorporated a quantitative aspect by presenting code frequencies in a systematic and tabular format (refer to Table 1). Methodological triangulation was employed, incorporating word cloud visualizations to enhance the interpretive depth of the findings.

### 4.1. Descriptive Analysis and Frequencies

All narratives included references to body-related risk factors or warning signs. The most frequently coded elements are presented in Table 1. Within the codebook of body-related risk factors, the most prominent themes were stress, exposure to violence, signs of trauma, lack of energy, self-harm, substance use, physical pain, and sleep disturbances. Regarding body-related warning signs, participants most frequently described symptoms directly associated with the suicidal mode [49].

The following section provides a detailed analysis of bodily-related risk factors and acute warning signs commonly reported across narratives. Given the interconnected nature of these experiences, some thematic overlap between the identified patterns is to be expected.

Beyond the body-related codebook (Table 1), additional thematic codebooks (Appendix A) were developed to capture other relevant aspects of the narratives. These codebooks included codes such as reported stressors, narrative styles, and descriptions of the suicide attempts. An alarming 17 out of 22 participants described a profound sense of not belonging, alongside equal reports of low self-esteem. Additionally, 16 out of 22 articulated feelings of being unheard or not taken seriously, alongside equal reports of pervasive hopelessness. Furthermore, a significant percentage (14 out of 22) traced the onset of their psychological distress back to childhood. In contrast, the remaining eight out of twenty-two participants reported that the stressors relevant to their suicide attempt emerged post-puberty. Among the spontaneously mentioned proximal triggers for suicide attempts, the most frequently reported factors included fear of the future (seven out of twenty-two), conflicts with peers (seven out of twenty-two), heartbreak (five out of twenty-two), and family conflicts (five out of twenty-two). Notably, five out of twenty-two participants explicitly stated that their suicide attempt occurred without a specific triggering event. This finding aligns with the fact that eight out of the twenty-two participants described the day of their suicide attempt as ordinary or even subjectively positive, suggesting that suicidal behavior is not necessarily tied to acute external stressors but may instead be linked to underlying persistent psychological distress.

### 4.2. Linguistic Analysis (Word Cloud)

Two types of word clouds were created: a single-word frequency cloud to highlight the most common individual words (Figure 1) and a two- to three-word combinations cloud to capture frequently co-occurring word pairs (Figure 2).

These visualizations serve as a starting point for understanding the dominant themes in the participants’ accounts and provide a structured yet intuitive way to explore how individuals articulate their physical distress, struggles, and perspectives on their suicide attempts. The linguistic analysis focused exclusively on body-related words to explore how physical experiences are represented in these narratives.

The results revealed that narratives frequently centered around substance use, self-harm, physical exhaustion, medical treatments, physical health, and experiences of violence and bullying. Additionally, when examining word clouds that consider two- to three-word combinations, it becomes clear that a negative body image—including issues such as body shaming, weight struggles, and eating disorders—plays a central role in the narratives of adolescents.

### 4.3. Content Analysis

In the following sections, we summarize the most frequent and clinically relevant categories emerging from the qualitative data. The following statement illustrates the complexity of triggers contributing to suicidal behavior:


*“I asked myself why I was always unlucky. I had nothing but bad luck during this time after the suicide attempt. I was bullied, I was harassed, I was robbed, I was touched even though I didn’t want to be. I have experienced so much adversity that I asked myself again why I am still alive and why I continue to endure this. I actually wanted to die. I simply should not have survived”.*
(Participant Z003)

This quote illustrates the interplay of various stressors and suggests that suicide attempts are not triggered by a single event but by a combination of stressors. Consequently, it is essential to further investigate stressors and triggers. In the following sections, body-related triggers and effects reported by patients are analyzed, and their relevance for the development of suicide attempts is highlighted.

#### 4.3.1. Violence and Traumatic Experiences

Given that a distressing 81.8% of included participants have experienced either violence or traumatic events or both, it is crucial to analyze these experiences and their physical consequences in greater detail. These experiences ranged from bullying and harassment to physical and sexual violence and were described as degrading and humiliating. For example, an adolescent described how other students pushed their head down into the toilet bowl and told them to drown themselves. Such experiences might have led to diminished self-esteem, guilt, and feelings of inadequacy. The resulting fears of being a burden often left victims suffering all alone, hiding their pain and feeling helpless and lonely.


*“But after one or two years, it wasn’t just light teasing anymore; it actually turned into bullying—with violence. […] I tried with all my energy not to show it at home, so that I wouldn’t be a burden. […] I used to lie and say I walked through some bushes, which is why I had scratches, and that I fell, which is why I was all bruised”.*
(Participant P001)

Such narratives illustrate how bodily marks anchor shame in the body and are associated with a sense of burdensomeness. This perception resonates with Joiner’s Interpersonal Theory of Suicide [36,37], which emphasizes perceived burdensomeness as a key factor in suicidal ideation.

Domestic violence, including physical and emotional abuse by parents, emerged as another dominant theme. One participant attempted to rationalize the violent parental behavior through cultural context or generational trauma yet still identified it as a key factor in their suicidal thoughts, which led to their first suicide attempt. The violence appears to have caused a sense of hopelessness and fostered a profoundly negative view of humanity and distorted expectations of relationships.


*“At home, I was beaten by my parents and also brutally bullied. ((…)) It was also emotional abuse. After an argument, they ignored me for two weeks. Afterward, they acted as if it didn’t happen. So, it’s very toxic. […] And then I thought to myself, if life goes on like this and people treat me this way, why should I continue living? (--) That same year, I also attempted suicide”.*
(Participant Z001)

Sexual violence was also present in several narratives and was often accompanied by internalized guilt and self-blame. The act of self-blame can be understood as a compensatory behavior, as victims often experienced feelings of loss of control, which led them to attribute responsibility to themselves rather than the perpetrator [64].


*“My older brother tried to rape or sexually harass me. I don’t want to go into detail or determine exactly what it was because one is a crime, and the other is not. I think it’s normal after such things, but I couldn’t sleep anymore out of fear that it would happen again. […] I didn’t wake up the next morning right away, and when I did, I couldn’t move. I was completely paralyzed. I couldn’t sleep at all for the next few days. I kept trying to wash the dirt off my body. My dislike for my body became much stronger because I felt like it was now dirty or something. ((…)) Since I considered myself responsible for raising my brothers, I took the blame for what happened”.*
(Participant P050)

This quote exemplifies the somatic imprint of trauma, where the physical body becomes a site of emotional contamination, leading to aversion and self-blame. The statement also emphasizes how the experienced trauma leads to maladaptive coping mechanisms, for example, compensatory behaviors, such as compulsive washing. Another adolescent’s reflection on how her “childish self” was “destroyed” through sexual violence highlights how the body was not just harmed but became the medium through which violation and objectification occurred.


*“And then I found out that I had been sexually harassed. A guy from my class took pictures of my butt and sent them around […] I used to sit next to him at school and he just took pictures of me and then texted strangers about raping me and that sort of things. And that completely destroyed my childish self. I was so little; I was 12 years old when that happened”.*
(Participant P043)

The last two quotes emphasize how the violation of personal boundaries and bodily autonomy can provoke emotional distress. Such experiences may contribute to stress-related physiological responses such as headaches, chronic pain, and fatigue [65]. Furthermore, the body is discussed as an object of harm that strips the affected person of agency, fundamentally altering their relationship with their own body.

Some participants described trauma-related dissociation, flashbacks, or paranoid perceptions, particularly following violent incidents. According to existing literature, traumatic experiences can lead to dissociation, suicidal ideation, and behavior [66]. Remarkably, the following participant is one of the rare participants who did not experience a suicidal mode but instead consciously decided to attempt suicide. They planned their suicide attempt in detail after stabbing someone in self-defense. After this traumatic experience, they developed persecutory beliefs and insomnia. The violent act triggered overwhelming feelings of remorse and guilt and ultimately led to a suicide attempt.


*“I started to feel like people were following me all the time, like at the train station. […] I got stuck in this thought: “What if I have killed someone?” […] That’s how things got to the point where I didn’t want to live anymore”.*
(Participant P035)

The quote emphasizes key concepts related to trauma, such as a heightened sense of alertness, unease in crowded spaces, intrusive thoughts, and flashbacks. Childhood traumas have been linked to neurobiological changes in brain functions, which can contribute to persistent somatic symptoms and heighten the risk for future health problems, including somatization, addictions, obesity, cancer, diabetes, and cardiovascular disease [67,68,69].

Medical and institutional interventions, such as physical restraint during hospitalization, were also experienced as traumatic and made the adolescents feel worthless, powerless, and helpless. The following quote underscores how the body becomes a site of vulnerability and distress, where control over one’s own body is stripped away:


*“I had tried to run away and was already restrained in the ambulance, they restrained me again in the hospital, tying me to a bed. I didn’t want any more help—I just wanted to die. […] At home, I kept having flashbacks of how the police restrained me, how I actually felt in my body”.*
(Participant Z001)

The adolescent’s focus on how they “actually felt” in their body highlights the profound physical and emotional impact of physical restraint. The inability to move in this case triggered a sense of hopelessness, whereby dying was a preferable alternative. As flashbacks are typically embodied experiences that can be vivid and overwhelming, the quote suggests that the body itself holds onto the experienced trauma [31]. This reinforces how bodily experiences are central to their narrative, shaping their understanding of the event and their sense of agency.

The following case illustrates how dissociation and a fight-or-flight response can be triggered by trauma-related experiences. As a result, physical aggression and overwhelming emotions were released.


*“Then I ran out (of the isolation room) because I couldn’t stand being in there, as I have trauma from the isolation room. The nurses ran after me. Then I got angry and threw a chair […] I just screamed and raged, shaking with anger. They stood in front of me and I couldn’t talk to them. I just shouted at them”.*
(Participant Z002)

In this case, the wish to run away seems to be immediately associated with being forced and losing agency and control over the situation. The trauma-related aggression appears to be a means of regaining control, which was often coupled with unmet emotional needs, such as the desire to have autonomy over their own body or to be protected. From this perspective, it seems plausible why traumatized individuals often develop compulsive behaviors after traumatic events [70].


*“Since then, I’ve had constant paranoia about being raped, which is why I’ve introduced almost obsessive control rituals, such as checking the door”.*
(Participant P050)

Despite seeking various forms of professional support, including hospitalization and therapy, many patients continue to struggle with the lasting effects of their traumas. The following quote illustrates how experienced trauma can lead to violent ideation.


*“I realized that he (the perpetrator) was doing similar things (sexual assault) to a friend of mine as well. So, we called the police again because we wanted to do something about it. And at some point, I realized that I had no control over it. Knowing that made everything worse because I couldn’t change it. And then I started having homicidal thoughts […] and I would think: “Oh, I’ll just take a knife to school and stab him”. My parents didn’t realize how serious I was about it”.*
(Participant P045)

The expressed homicidal thoughts are an expression of a profound sense of powerlessness, injustice, and rage. These thoughts could potentially lead to impulsive decisions to regain some control. Especially, as the adolescent also reports suicidal tendencies, the risk of self-harm or harming others seems to be elevated, as they may perceive that they have nothing left to lose. Additionally, the adolescent describes the feeling of not being taken seriously by her parents. The perceived invalidation may intensify feelings of helplessness, loneliness, and social isolation.

In summary, these findings highlight how violence and trauma alter adolescents’ relationship to their own bodies, transforming them into objects of self-blame, lack of control, helplessness, hopelessness, and shame. Perceived burdensomeness and Thwarted Belongingness, as framed by the Interpersonal Theory of Suicide [37], emerged as important factors in understanding the interplay between trauma and suicidal behavior. In this sense, the body becomes a battleground where emotional pain is internalized and psychological injuries manifest physically.

#### 4.3.2. Body Shaming and Body Dissatisfaction

The second very prominent topic in the narratives of adolescents was body dissatisfaction, which was often fostered by social comparisons, body shaming, or unintentional weight increases. Approximately half of the patients spontaneously mentioned that they had either felt uncomfortable with their bodies or had developed an eating disorder, which is associated with body dissatisfaction. Several participants linked the onset of body dissatisfaction to physical development during puberty, particularly to an early onset of puberty.


*“I also went through puberty earlier. That was in the 4th grade, while others were still so little, and I was already growing. I got wider hips and all that. The girls commented, “She’s wearing a bra” or “Ugh, she’s got her period”. And the boys laughed at my breasts. […] I never really liked my body. Then, during the quarantine (COVID-19), I gained weight. I was around 65 kg, and that was more on the overweight side, as my doctor confirmed. […] I didn’t like my body, and I also hid myself in big clothes and so on. At some point—at the end of September—I just stopped eating consciously. […] I lived almost exclusively on cornflakes and Diet Coke. […] In the end, I was even underweight”.*
(Participant P049)

This quote aligns with research findings suggesting that early puberty is considered a risk factor for negative self-perception and maladaptive coping strategies [71,72]. The example illustrates how the adolescent became more aware of their body and how weight concerns led first to feelings of shame and self-hatred, and later to the development of an eating disorder and a suicide attempt. At the same time, this quote addresses several body-related mechanisms of action that could have contributed to the suicide attempts, such as body shaming, social comparison, and peer pressure. Remarkably, participants who also showed self-esteem issues seemed to cope with events that triggered body dissatisfaction in a maladaptive way, such as by developing eating disorders. This correlates with the findings of Fan et al. [73], which showed that a lack of self-compassion moderated the interplay between body dissatisfaction and suicidal ideation.

Furthermore, the participant mentioned the attempt to hide their body under oversized clothes, indicating a desire to mask their dissatisfaction. In this sense, the body can be seen as an externalized battleground of inner struggles, in which the participant is afraid of showing their vulnerability to others and being evaluated by them. The desire to hide or to take up less space can also be considered an indirect form of suicidal behavior, as one’s own body is battled through conscious restrictive eating.

Additionally, participants reported engaging in extreme dieting or disordered eating behaviors during the COVID-19 lockdown, including weight gain due to binge eating, excessive consumption of sugary drinks, or physical inactivity. Due to social isolation, the appearance of the body seemed of secondary importance. The end of the COVID-19 lockdown and, consequently, the return to social life required exposure to bodily transformation, which was associated with a great sense of shame for the participants. Similarly, according to another study [74], the impact of the COVID-19 lockdown led to lasting physical changes, such as eating disorders and obesity, which increased reports of anxiety, depression, and suicide attempts. In another study, a direct association between body dissatisfaction and physical inactivity was found, which is related to suicidal ideation [75].

Experiences of body shaming by peers emerged, as previously mentioned, as a prominent topic in the narratives of adolescents. However, not only does non-normative body shape lead to body shaming but also other attributes of appearance—such as scars, nose shape, and skin color. Adolescents reported being labeled as “hippo”, “anorexic girl”, or as a “stick”. These expressions of bullying suggest that body shaming is not necessarily tied to an overarching beauty ideal. Instead, adolescents may experience a sense of hopelessness, as others may always find a beauty flaw. Societal and medical influences were also frequently cited as contributors to unrealistic beauty standards.


*“I just wanted to lose weight around my stomach. I saw it on the internet or in a fashion magazine and was like ’Wow, the model had a really nice flat stomach’. I knew that it was photoshopped or fake, but it still subconsciously influenced me to try to lose weight, and it just got worse and worse, so I got more and more restrictive”.*
(Participant P044)

Even when the digital manipulation of social media content was recognized by the participants, they appeared to internalize these ideals, which often resulted in eating disorders. This could be explained by the distorted body perception, which is associated with eating disorders. Some participants also reported how social validation of weight loss reinforced restrictive eating, which fostered the vicious cycle of weight loss. Notably, while all the female participants tended to report feeling “too fat”, some male participants also reported feeling uncomfortable in their bodies because they considered themselves too thin or not muscular enough.

Surprisingly, some patients experienced body shaming and bullying not only from peers but also from adults (parents, teachers, doctors). Particularly, parents were reported often to criticize their children’s weight and appearance. This parental influence significantly negatively impacted one’s body perception and satisfaction. Affected participants who experienced body criticism in childhood reported later having deep-rooted self-worth issues and eating disorders. The following quote highlights how the participant was made aware of her body image as a child by their parents’ critical comments about their body. Later, their weight was tracked and shamed. In line with their parents’ view, they developed a distorted body image, started counting calories, eating restrictively, and vomiting.


*“I was only five when I realized that my parents were talking critically about my sister’s body. […] That made an impression, and I memorized that calories are associated with something bad. […] My parents started body-shaming me when I was 11. When I was 12, it got really bad, and I had to get on the scales in front of my mum, and she said I already weighed more than her […] I started eating less and vomiting”.*
(Participant Z005)

In this case, it seems as if the participant had classified their body as inadequate due to family expectations. Combined with the adolescent’s perfectionist personality, the feelings of inadequacy led to the perception of the body as an enemy and a cause of suffering. The resulting eating disorder could, therefore, be seen here as a form of self-destruction—in favor of self-punishment for not being good enough. Even in the longer term and despite various therapeutic interventions, she has not yet managed to overcome the eating disorder.

In summary, the role of body dissatisfaction in narratives of adolescents with attempted suicide appears to be significant. Mainly, body criticism, such as body shaming and social comparisons, seems to contribute to body dissatisfaction. In addition, an early onset of puberty and the COVID-19 lockdown have further had a negative impact on body satisfaction.

#### 4.3.3. Body-Damaging Coping Mechanisms

As a consequence of body dissatisfaction and exposure to violence or traumatic experiences, participants often reported body-damaging behaviors and maladaptive coping mechanisms, such as self-harm, substance use, eating disorders, and suicidal behavior. These destructive behaviors toward their bodies can be interpreted as an expression of internal struggle. The body, which is often perceived as a source of suffering, is deliberately harmed—partly as a way of self-punishment or to regain control. Ultimately, the body is used as a projection of psychological distress to cope with deep pain and self-hatred.

**(a)** 
**Self-harm**


Many participants reported that they began self-harming during their early adolescence, particularly in the context of a depressive episode. Self-harm was initially perceived as a harmless and controlled behavior but frequently escalated over time, particularly in connection with a growing sense of indifference toward life. Accordingly, distinguishing between self-harm and suicide attempts was not always easy.


*“I cut my neck once and thought, if it happens, it happens, and if not, then not. I did it several times, and I did the same to my arm, at the pulse artery”.*
(Participant Z005)

The following quotes demonstrate how the body becomes a battleground in the context of self-harm to compensate for emotional pain, such as shame, feelings of worthlessness, self-hatred, and the unmet need for control. Some participants used self-harm as a form of self-punishment, pointing to a negative perception of the body.


*“I had to hurt myself because I wanted to see myself bleed or feel pain. But also, because I hate myself and I deserve it”.*
(Participant P047)

Beyond self-hatred, the participant reports a need to “feel pain”, which indicates a further function of affirming one’s own existence. Other participants framed self-harm as a substitute for suicide. Self-harm is used here as both a survival strategy to regulate suicidal ideation and as an expression of feeling unworthy. In several cases, self-harm became compulsive and addictive, with visible injuries serving as a cry for help. Two key commonalities emerged among those affected participants: low self-esteem and difficulty in managing overwhelming emotions. These findings align with the existing literature, which understands self-harming behavior as a behavioral addiction [76] and proposes to treat self-harm as an addiction [77].


*“I called for help. Every time I went to the toilet, the floor was covered in blood, at home, and at school […] Because when I cut myself, I see how it opens up, gapes, and the floor is covered in blood—that relaxes me […] I think it’s cool to show that, it’s actually similar to when you have some kind of addiction—you always have the urge to do it”.*
(Participant Z001)

Beyond its addictive effect, the quote reveals how a bleeding body can bring temporary relief from emotional pain into physical pain. The body is instrumentalized and harmed to control and regulate emotional pain. In addition, the emotional pain is symbolically visualized through scars, which the participant even perceives as “cool”. In this case, the wounds are used as a communication tool to externalize internal suffering. The scars were often used as a means of seeking recognition, whether as a call for help at school, in a hospital, or at home. The public display of physical wounds was often “rewarded”, as many patients reported that they first received therapeutic support after making their self-harm visible in public.

**(b)** 
**Substance Use**


The second body-damaging coping mechanism revealed by the qualitative analysis was substance use. Several patients described using substances as a maladaptive coping strategy to numb emotional pain, seek pleasure, or escape from reality. Cannabis was often used to suppress anxiety, induce emotional detachment, or facilitate sleep. Despite its perceived short-term benefits, many adolescents experienced an exacerbation of their psychological problems in the long term, often leading to addiction, weight gain, or impaired academic performance. Once again, the body emerged as a battleground between avoidance, self-hatred, self-destruction, and suicidal ideation.


*“I developed sleep problems and could only sleep three hours a night, so I started smoking weed to cope […] Smoking weed also led to binge eating, causing me to gain 18 kg, which made me feel extremely uncomfortable in my own body […] I used cannabis to escape reality. Eventually, I realized I had to stop because my school performance was deteriorating—my grades kept worsening”.*
(Participant Z013)

While most of the participants recognized the need to discontinue substance use to achieve positive life changes, others remained convinced that numbing the body through substances was beneficial for managing distress. For instance, one patient reported the following:


*“My only medication is smoking weed, which I actually do every day now. I can sleep better, and I’m in a world where nothing bothers me, and all the problems feel less drastic. It’s much more relaxed, and I don’t always have to think about what’s wrong”.*
(Participant P043)

This participant uses cannabis as a form of self-medication and attempts to cope with psychological symptoms while trivializing the long-term negative consequences. Such perceptions may be reinforced as cannabis is often perceived as a “harmless” substance. However, the meta-analysis by Gobbi et al. [78] indicates that non-medical cannabis use is associated with an increased risk for suicidal ideation and behavior. In addition, many patients reported that they initially experimented with substances such as alcohol, cannabis, medication, and nicotine—either out of curiosity or under peer pressure—which later escalated into addictive use, also of more potent substances. The fight against the own body, therefore, does not only take place isolated but also often within social groups. As suicidal adolescents often experience a missing sense of belongingness in line with Joiner [36,37], adolescents may gain social validation through social substance use, further reinforcing addiction.


*“I started using alcohol and cannabis when I was 16. I moved to stronger drugs like LSD later. I used them weekly, and at some point, I even started drinking at school. I had the urge to do it, often with friends who also felt this urge”.*
(Participant Z003)

As highlighted in the quote, the progressive substance use escalated beyond control. The following quote illustrates the extent to which substance use resulted in physical harm (e.g., nasal injuries) and engagement in high-risk behaviors. In this case, the physical damage was accepted as a consequence of the inner fight.


*“I started with marijuana and progressed to cocaine, MDMA, and LSD. Simply to feel some form of pleasure. That went on for quite a long time, and at some point, I got nasal injuries […] During my last MDMA trip, I hallucinated about my dead grandfather—3 pills at once. I ended up on a bridge, and my friend had to restrain me because I was already leaning over the railing”.*
(Participant Z034)

Here, the participant consumed three highly dosed pills of MDMA without any initial suicidal intent. In general, it is alarming how frequently suicide attempts among participants occurred impulsively under the influence of substances, particularly alcohol. The combination of pre-existing indifference towards one’s own life and substance-induced hallucinations often led to acutely life-threatening situations.

**(c)** 
**Eating Disorders**


The third body-damaging coping mechanism identified by the qualitative analysis was the manifestation of eating disorders, often rooted in a desire to control their body weight. Participants frequently described a self-destructive cycle that led to restricted eating, vomiting, purging, or excessive exercising, with some even requiring feeding tubes. Furthermore, several examples revealed that the fear of gaining weight remained an expressed burden, even when the adolescents’ weights were within or below the normal range. This phenomenon indicates a distorted body image, a central component of anorexia nervosa [79], which in turn is also associated with one of the highest suicide rates among psychiatric disorders [80].


*“I was so afraid of gaining weight that I would vomit after binge eating. I began to develop brutal strategies. If I ate too much, I would vomit. If I did something bad, I would hurt myself. And if I did something wrong, I wanted to kill myself. Those were always my solutions”.*
(Participant P049)

This distorted perception of body image further exacerbates the use of maladaptive coping mechanisms, such as vomiting or suicidal behavior, which may be interpreted as a form of self-punishment. Particularly among perfectionistic participants, weight fluctuations and the fear of weight gain were revealed as immediate triggers for suicide attempts. This suggests that perceived loss of weight control and feelings of defeat experienced among this group must be considered significant risk factors for suicidal behavior.


*“I managed to control myself for quite a long time. I would describe myself as very ambitious and a perfectionist, which I maintained for a long time. I ate nothing but corn waffles and similar stuff for days and months”.*
(Participant P049)

In extreme cases, eating disorders acted as an “indirect suicide plan” as a last resort in the participants’ fight for control, and suicide attempts were directly triggered by the perceived weight gain. The imperfect body image became intolerable to the extent that this participant preferred death. The perceived loss of weight control was interpreted as a failure, ultimately leading to attempted suicide.


*“I noticed that I had gained weight. My plan had failed. I gained 6 kg, and that’s quite a lot for my anorexia. I went home and decided to take painkillers. That was the first thing that came to my mind because I couldn’t control my eating. I felt hungry and I couldn’t resist. And then I swallowed eight 500 mg tablets […] Then, I spent a night in the intensive care unit. I hoped to stay there because I don’t know why, and it may sound sick, but I liked being there. I was with the supervisor, and I got full attention, and getting attention has always been a great need of mine”.*
(Participant P044)

The participant expresses a pronounced need for attention and validation. The intensive care unit was regarded as favorable, as the participant felt seen and cared for. This need reveals underlying emotions such as loneliness, a sense of not being taken seriously, and insufficient social support. The resulting illness identity was shaped by secondary gains (attention, care), which is often referred to as Hospitalism or Institutionalization [81]. Illness identities are associated with psychological maladjustment and feelings of shame and self-criticism, which may lead to dysfunctional behavior. Further, as the impulsive suicidal action was expressed as a response to “irresistible hunger”, a battle between bodily needs and distorted beliefs about food was uncovered. Once more, physical needs lost the battle against the benefit of maladaptive psychological beliefs.

#### 4.3.4. Acute Physical Warning Signs

In addition to long-term body-related risk factors, many suicidal participants reported acute physical warning signs preceding or during suicide attempts, including a profound sense of loss of control and a sense of bodily disconnection. Many participants described dissociative states, including derealization and depersonalization, as prominent phenomena during suicide attempts. These states were often characterized by a trance-like state, wherein several participants experienced as if they were no longer fully inhabiting their bodies or perceiving them accurately. Some described numbing sensations, expressing a dissociation from their actions and emotions, accompanied by an absence of physical pain.


*“I was completely gone; I don’t know what I did then. I had already been sitting there for two hours, and at some point, I asked, ‘What did I actually do?’ […] Then I suddenly noticed that my shoe was wet and that I had torn out my nail. I was in a suicidal mode. But somehow, it was almost robotic. I saw the rope, I saw the table, and I thought, “Oh, I could hang myself”—and just did it”.*
(Participant P001)

The loss of control was frequently described as entering a “robot-mode”, where participants felt as if they were “remotely controlled” or as if an irreversible “switch” had been activated. These experiences align with the literature on the suicidal mode, which emerges as a coping mechanism for stressful experiences [49].


*“I thought I was not needed at all. Basically, I am just a burden to the people around me. And I was like, okay, I don’t want this anymore. I’m tired […] A switch had flipped inside me, and I had no desire anymore. I didn’t want to go to school or anywhere else—I just wanted to leave and took the pills”.*
(Participant P032)

In this case, the participant’s suicidal mode was triggered by an adverse personal experience, a dispute with a close friend shortly before the suicide attempt. The perceived burdensomeness may have contributed to emotional pain, leading to a loss of control and an impulsive suicidal action. Similar experiences were expressed by others, who felt as if they were observing their actions from the outside. Given this lack of awareness, it is not surprising that most patients reported memory gaps surrounding their suicide attempts.

Extreme exhaustion (tiredness) was also revealed as a prominent warning sign, aligning with markers of depressive episodes. Many participants described feeling physically drained, tense, and overwhelmed, often spending long periods isolated, particularly in bed. The physical exhaustion frequently resulted in social withdrawal and an inability to maintain a regular daily routine. The lack of daily structure, partly exacerbated by the COVID-19 lockdown, contributed to increased inactivity and boredom, leading to feelings of anhedonia, hopelessness, and diminished quality of life. Interestingly, not all participants experienced physical exhaustion. Instead, a subgroup reported insomnia and experienced a temporary increase in productivity and positive mood.

Additional physical warning signs frequently reported during suicide attempts included dizziness, tremors, panic attacks, cold sensations, tachycardia, shortness of breath, hearing loss, and blurred vision. In the subacute phase, during the last hours and days before the suicide attempt, numerous participants retrospectively identified physical warning signs such as loss of appetite, sleep disturbances, physical discomfort, and pain (particularly headaches and abdominal pain). Many also reported heightened tension and extreme lethargy. Some participants described speech impairments, struggling to formulate complete sentences, and resorting to minimal verbal responses, such as “yes” and “no”. This speech pattern further contributed to their social withdrawal. Notably, several participants reported neglecting their personal well being and bodily needs, reflecting their inner battle. This neglect manifested in areas such as a decline in personal hygiene, irregular eating, and a general disregard for their own needs. The bodily disengagement highlights how the body became a battleground of both emotional suffering and physical disconnection.

## 5. Discussion

### 5.1. Summary

This paper provides deep insights into the complex relationships between body-related experiences, physical responses, and their impact on mental health through the qualitative analysis of narrative interviews with adolescent patients who attempted suicide. Furthermore, some quantitative aspects served to structure and support the interpretation of narrative content. The findings emphasize the central role of the body in adolescent suicidal behavior, revealing two primary body-related risk factors: (1) Experiences of Violence or Trauma and (2) Body Dissatisfaction. The presence of these two factors appears to contribute to the engagement in body-damaging coping strategies, including self-harm, substance use, and eating disorders.

Based on the qualitative analysis of the narratives, it is hypothesized that the interplay is moderated by a range of psychological constructs, such as self-blame, shame, self-hatred, lack of control, helplessness, hopelessness, and perceived burdensomeness. Consequently, adolescents may find themselves entering suicidal modes, characterized by experiences of dissociation and a sense of loss of control, highlighting the embodied nature of suicidal crises.


The Influence of Experienced Violence and Trauma


The findings indicate that adolescents who have experienced violence or trauma develop a disrupted sense of basic trust. Feelings of powerlessness during traumatic events appear to contribute to a belief that they have no control over their lives, feeding perceptions of constant unpredictability and danger. In instances where emotions of entrapment were also triggered by intrusive memories or flashbacks, suicidal behavior often emerged as an attempt to escape unbearable emotional pain and the risk of further traumatic experiences. In addition, memories of traumatic experiences can also contribute to dissociative responses, such as flashbacks, which may contribute to impulsive suicide attempts. At this point, some adolescents engaged in reflections on sociopolitical issues, questioning the value of life in a world they perceived as brutal and unjust (e.g., war, rape, bullying). However, experiences of violence not only disrupted the need for safety and control but also self-esteem. Many adolescents reported intense feelings of shame, humiliation, and self-blame, often internalizing their responsibility for the traumatic event. The inability to defend themselves contributed to deep-seated self-hatred and anger. While self-blame appeared to intensify feelings of inadequacy, the overwhelming feelings of guilt and remorse often resulted in a diminished desire to continue living. These findings are consistent with trauma theories that emphasize the embodied nature of psychological suffering [31,32]. The adolescents’ narratives indicate that the body is not merely affected by psychological pain, it becomes the site through which it is experienced and expressed. In summary, the findings highlight the urgent need for targeted interventions after experiencing trauma or violence. However, not only psychological support is essential; physical healing also needs to be considered to recover from the traumatic past.


The Role of Body Dissatisfaction


Body dissatisfaction emerged as a serious psychological burden for adolescents who had attempted suicide, frequently resulting in diminished self-esteem. Various factors contributed to body dissatisfaction, including experiences of body shaming, bullying, sexual assault, early onset of puberty, weight gain, and social comparisons—particularly in the context of social media. Both overweight and underweight adolescents experienced body shaming, emphasizing that perceived physical inadequacy, rather than objective body shape, was the key issue.

The findings of this study align with research on the psychological effects of digital media exposure. In today’s digital environment, these pressures are amplified by social media platforms that reward visibility, comparison, and idealized self-presentation [20]. As prior studies have shown, exposure to unrealistic body standards and cyberbullying can significantly increase body dissatisfaction and suicidal ideation [23,24,25]. Our findings reflect this dynamic, as for many participants, the body became a medium of public performance through which self-worth and social belonging were negotiated.

The findings revealed how the body serves as both a projection of social and familial expectations and as a reflection of inner struggles and psychological burdens. These findings align with studies that state that body dissatisfaction can predict the onset of depression and is considered a risk factor for suicidal ideation [73,82]. These insights are further supported by the findings of a meta-analysis by Linardon et al. [83], showing that body appreciation is positively associated with various aspects of psychological well being, including self-esteem, self-compassion, and sexual satisfaction. Since the interplay between body satisfaction and suicidal behavior appears to be relevant, the development of psychoeducational interventions aimed at improving body satisfaction is essential. These interventions should promote self-compassion and body acceptance. However, as the perception of one’s body is also influenced by external criticism, family counseling and educational campaigns at school and among healthcare professionals are required, as well as interventions designed to promote resilience against social pressures.


Body-damaging Consequences and Coping Mechanisms


The findings indicate that adolescents in suicidal crises use their bodies as battlegrounds for psychological pain. Experiences of trauma, violence, and body dissatisfaction contribute to maladaptive and body-damaging coping mechanisms. Participants frequently reported engaging in self-destructive coping mechanisms such as self-harm, substance use, eating disorders, or suicidal behavior. These behaviors appear to reflect attempts to regain control, self-punish, or externalize psychological distress. A common characteristic of adolescents who harmed their bodies was an unmet need for attention and validation.Self-harm was frequently described as a coping strategy providing temporary relief and a means of managing self-hatred. Several adolescents reported that displaying their scars of self-harm was the only way their distress was recognized, which eventually led to supportive interventions.Substance use was often used to numb emotional pain or escape from reality, which in some cases, led to impulsive suicide attempts under the influence of substances. Given the fragile mental health of these participants, it is not surprising that some developed substance use disorders that further impaired their ability to regulate emotions and behavior. The co-occurrence of suicidal ideation and substance use constitutes a particularly high-risk constellation, as the disinhibiting effects of substances can lower the threshold for impulsive suicide attempts and attenuate the innate fear of death, thereby further compromising internal protective barriers and increasing the likelihood of suicidal behaviors.Eating disorders frequently served as a means to manage body dissatisfaction or to fulfill a desire for control. The battle against hunger often resulted in perilous cycles of self-punishment and feelings of inadequacy, ultimately including suicidal acts. The significance of eating disorders aligns with the findings of a meta-analysis conducted by Amiri and Khan [29], which revealed an interplay between eating disorders and suicidal behavior, while suicidal ideation was prevalent in 60% of individuals with eating disorders, and suicide attempts were reported in 25% of this population.

As the usage of the mentioned maladaptive coping mechanisms was, in some instances, rewarded with attention (a secondary gain), self-destructive behaviors may have been reinforced, potentially contributing to the maintenance of an illness identity (hospitalism, institutionalization) by further damaging the body. Finally, adolescents appear willing to pay the price of physical damage in exchange for psychological needs. This willingness to sacrifice the own body highlights the urgent need for targeted interventions promoting bodily self-care and adaptive coping strategies.


Acute Physical Warning Signs


The concluding important finding of this paper relates to the assessment of acute somatic warning signs in suicidal behavior. The analysis demonstrates that the body plays a crucial role not only in long-term patterns but also in immediate contexts of suicide attempts. Participants frequently reported experiencing profound dissociative states during their suicide attempts, characterized by a sense of detachment from their bodies and a loss of control. These states were frequently described as being in “robot mode” or a “trance-like state”, suggesting significant psychophysiological dysregulation. Further physical warning signs, such as extreme exhaustion, neglect of bodily hygiene, and speech impairments, further emphasize the central role of the body as a battleground for psychological pain. These findings underscore the necessity of further investigating the role of dissociation in the escalation of suicidal behavior and show the need to address both the emotional and physical aspects of suicide prevention.

### 5.2. Strengths and Limitations

A strength of the present study is its utilization of narrative interviews with adolescents who have a history of suicide attempts. As these narratives are raw and unfiltered first-person accounts, they provide an authentic and direct insight into the subjective experiences and triggers of suicidal behavior. This approach allows for a deeper understanding of the psychological mechanisms underlying suicidal behavior. Another notable strength is the innovative focus on body-based experiences, which, to the best of our knowledge, is an underexplored area in previous research. By highlighting the role of body-based experiences in suicidal ideation and behavior, the findings may inform the development of more effective interventions aimed at the early identification and support of vulnerable individuals. Furthermore, integrating qualitative narrative analysis with quantitative assessments of linguistic and descriptive features increases the robustness of this study and enables a more comprehensive examination of language and frequency patterns. This methodological triangulation validates qualitative findings and provides insights into the prevalence and significance of specific sources of stress and recurring topics. This study highlights key areas that should be prioritized in targeted intervention approaches by identifying the subjectively relevant topics.

Despite its strengths, the present study encompasses several limitations that offer important implications and directions for future research. First, the data are based on retrospective self-reports, which may be influenced by recall bias. Second, the sample was drawn from a specific clinical population—adolescents who have attempted suicide—potentially limiting the generalizability of the findings. Even within this clinical population, generalizability remains questionable, as this study exclusively included patients who voluntarily participated in a short intervention program (AdoASSIP) for the secondary preventive treatment after a suicide attempt. This implies that the sample consists of relatively stable adolescents willing to live. Another limitation concerns the gender distribution of the sample, which was not representative of the general adolescent population. However, the gender distribution in the AdoASSIP intervention program was similarly unrepresentative, which aligns with the gender paradox of suicidal behavior, wherein women tend to attempt suicide more often, but men are more likely to die by suicide [84].

Another limitation pertains to language, as the interviews were conducted in German or Swiss German and subsequently translated into English. This translation process may have resulted in the loss of subtle nuances in language and cultural context. Finally, for the descriptive analysis, it must be acknowledged that the sample size of n = 22 is insufficient. Nevertheless, for the qualitative study, this sample size is considered substantial [85,86].

### 5.3. Implications and Future Research

Understanding the body as a battleground in the context of suicidal behavior offers critical insights for the development of effective prevention and intervention strategies, highlighting the necessity for holistic approaches that encompass both psychological and somatic dimensions of distress. Traditionally, suicidal behavior research has focused on emotional and cognitive processes, often overlooking the profound impact of body-related distress. Our findings highlight the necessity to shift this paradigm, recognizing that body-related risk factors and warning signs are significant in detecting acute suicidal ideation and preventing suicide attempts.

Experiences of violence, trauma, and body shaming emerged as alarming risk factors. Furthermore, body-damaging coping mechanisms, including eating disorders, self-harm, and substance use, could serve as important indicators that the individual may be in a state where the body serves as a battlefield of suicidality. Acute warning signs, such as the experience of the suicidal mode, characterized by a dissociative state, require heightened attention during risk assessments. Given that the suicidal mode could be re-triggered by subsequent stressful experiences [87], it is essential to implement targeted interventions post-suicide attempt. These interventions should aim to raise awareness of potential warning signs and to acknowledge the persistently high suicide risk of reactivation of the suicidal mode and support both the affected adolescents and their support systems, improving their ability to recognize early warning signs and effectively implement emergency plans. The AdoASSIP [48] presents a promising intervention in addressing these concerns. It is designed to enhance awareness of adolescents and their support systems by providing insights into their experiences and perspectives, thereby highlighting unmet needs while actively involving the support system in implementing the emergency plan. Nevertheless, there is an urgent need to focus more intently on body-related aspects within the context of suicidal ideation, as these factors can yield essential information regarding the adolescent’s state.

To advance our understanding, future studies should extend their focus beyond emotional and cognitive processes to explore the complex interactions between physiological and psychological mechanisms. Increasing the sample size would enhance the representativeness and generalizability of the findings. Additionally, longitudinal research designs could help to establish causal relationships, thus supporting the development of evidence-based prevention and intervention programs. Incorporating both quantitative and qualitative approaches is essential to further investigate the adolescents’ perceptions of stressors and identify specific triggers for suicide attempts as reported in narrative interviews. This would also provide deeper insights into potential variations in suicide risk and coping mechanisms.

Therefore, a comprehensive, interdisciplinary approach to suicide prevention should be used, incorporating clinical, psychological, and sociopolitical strategies to empower adolescents with functional coping mechanisms. By addressing underlying vulnerabilities interprofessionally, these efforts may ultimately reduce the overwhelming sense of distress that often precedes suicidal behavior and enhance the resilience of vulnerable adolescents. From a clinical perspective, greater emphasis should be placed on addressing body-related distress through targeted interventions that promote a positive body image, strengthen self-efficacy, and enhance social support systems. Integrating body-oriented therapeutic approaches, such as relaxation techniques and mindfulness-based interventions, could be particularly effective in reducing perceived stress among adolescents. Given the powerful influence of the internet and social media on adolescent body image and self-worth, future prevention and intervention strategies should consider incorporating guidelines for healthier media use. Emerging findings suggest that reducing social media exposure may mitigate body dissatisfaction and related psychological distress. From a sociopolitical perspective, it is, therefore, essential to examine the extent to which unrealistic beauty standards contribute to adolescent distress and to develop preventive strategies to reduce these stressors. Beyond this, machine learning techniques, such as Natural Language Processing (e.g., sentiment analysis), could refine data analysis, uncover additional patterns in narrative reports, and improve predictive modeling of suicide risk.

Finally, the unexpectedly high prevalence of trauma-related symptoms (63%) compared to diagnosed PTSD (18%) raises critical questions regarding potential underdiagnosis, which may hinder access to appropriate, disorder-specific treatments. While narrative accounts provide valuable subjective insights, they are not a substitute for clinical diagnosis. Nevertheless, this discrepancy highlights the necessity for improved clinical identification of trauma-related symptoms. It is plausible that not all individuals exhibiting PTSD-like symptoms meet the full diagnostic criteria for PTSD. However, the presence of clinically relevant trauma symptoms should not be overlooked, as timely and adequate treatment may serve as a crucial protective factor against suicidal behavior. This also touches upon a broader sociopolitical issue. In many healthcare systems, access to evidence-based interventions is often tied to formal diagnoses, potentially excluding individuals in significant psychological distress from receiving necessary support.

In summary, body-related factors remain an underestimated dimension in suicidality research. Future research should prioritize a more comprehensive integration of body-related elements and provide essential insights into adolescents’ perceptions to develop more effective risk assessment strategies and intervention approaches.

## 6. Conclusions

The findings underscore the necessity to consider the body as a battleground for suicidal behavior, emphasizing the critical influence of experiences of violence or trauma, and body dissatisfaction as key contributors to adolescent suicidal behavior. These factors appear to promote an unhealthy lifestyle and maladaptive coping strategies, such as self-harm, substance use, and eating disorders, often intensified by problematic media use and exposure to harmful online content. Moreover, emotional and cognitive moderators—including self-hatred, shame, guilt, and feelings of inadequacy—further exacerbate vulnerability, potentially leading to psychophysiological dysregulation and dissociative states, which are closely associated with increased suicide risk.

Finally, on the battlefield of adolescents in crisis, the body becomes both a shield, a weapon, and a site of self-destruction. Moreover, the body is a source of shame, a victim of violence, an instrument of self-destruction, and the ultimate site of emotional expression. This complicated interplay requires a radical change in our approach to suicidal behavior, moving beyond traditional analysis of risk factors. Further, the complex interplay urges us to listen more carefully to the narratives of those affected and to give more relevance to the body-related aspects, which can be interpreted as the tangible reality of emotional pain. These nuances and details may hold the most important clues to suicide prevention, leading to the development of life-saving interventions.

## Figures and Tables

**Figure 1 ijerph-22-00859-f001:**
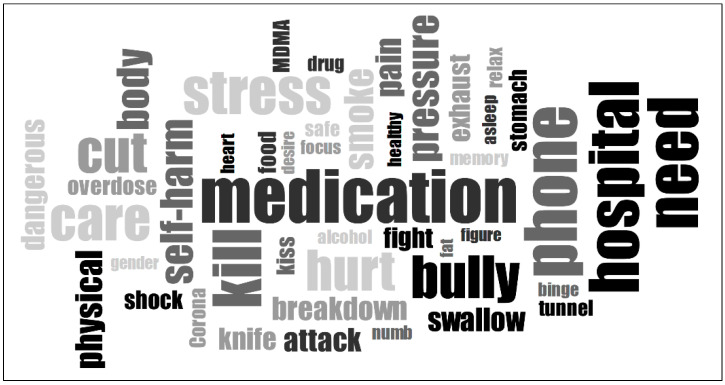
Single-word cloud of body-related language in suicidal narratives. The frequency of each word’s occurrence is reflected in the size of the word in the cloud. Different shades of black are used for visual variety only and do not convey any additional meaning.

**Figure 2 ijerph-22-00859-f002:**
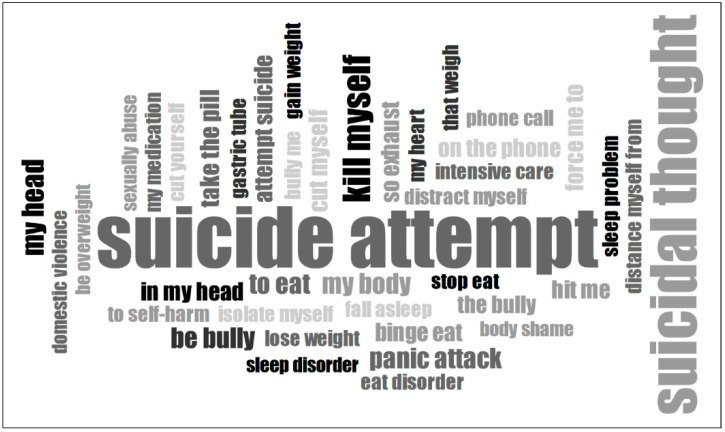
Two- to three-word cloud of body-related language in suicidal narratives. The frequency of the word’s occurrence is reflected in the size of the word in the cloud. Different shades of black are used for visual variety only and do not convey any additional meaning.

**Table 1 ijerph-22-00859-t001:** Codebook “Body as a Battleground” and overall frequency of codes.

Code	Participants with Code (*n*/N)	Overall Frequency of Code (N)
1. Body Shaming	50% (*11*/22)	21
1a. Body Dissatisfaction	31.8% (*7*/22)	12
1b. Experiences of Body Shaming by Adults	18.2% (*4*/22)	5
1c. Experiences of Body Shaming by Peers	31.8% (*7*/22)	8
2. Violence	22.7% (*5*/22)	85
2a. Verbal Abuse and Bullying	72.7% (*16*/22)	42
2b. Violence by Peers	45.5% (*10*/22)	10
2c. Violence by Adults	27.3% (*6*/22)	13
2d. Sexual Violence (e.g., Sexual Abuse)	22.7% (*5*/22)	13
3. Signs of Trauma (e.g., Flashbacks)	59.1% (*13*/22)	33
4. Self-harm	72.7% (*16*/22)	45
5. Perceived Stress	86.4% (*19*/22)	119
5a. Overburdening Responsibility	40.9% (*9*/22)	13
5b. Performance Pressure	81.8% (*18*/22)	51
6. Sleep Disorders	50.0% (*11*/22)	23
7. Eating Disorders	45.5% (*10*/22)	59
7a. Stress from Weight Gain	22.7% (*5*/22)	7
7b. Fear of Gaining Weight/Calories	18.2% (*4*/22)	6
7c. Distorted Body Image	13.6% (*3*/22)	3
7d. Binge Eating	13.6% (*3*/22)	4
7e. Purging Behaviors	22.7% (*5*/22)	7
7f. Restrictive Eating/Anorexia	36.4% (*8*/22)	17
8. Substance Use and Addiction	63.6% (*14*/22)	59
8a. Alcohol Use	18.2% (*4*/22)	5
8b. Cannabis Use	22.7% (*5*/22)	13
8c. Tobacco Use	22.7% (*5*/22)	6
8d. Harder Substance Use	13.6% (*3*/22)	9
8e. Problematic Media Use	40.9% (*9*/22)	11
8f. Addiction	22.7% (*5*/22)	8
9. Lack of Motivation/Energy	77.3% (*17*/22)	80
9a. Lack of Physical Activity	31.8% (*7*/22)	14
9b. Absenteeism and Lack of Structure	40.9% (*9*/22)	18
9c. Neglect of Self-Care (e.g., Body Hygiene)	18.2% (*4*/22)	6
10. Acute Physical Symptoms during Suicide Attempt	90.9% (*20*/22)	110
10a. Suicidal Mode (Physical Aspects only)	86.4% (*19*/22)	70
10b. Somatic Pain or Discomfort	27.3% (*6*/22)	7
10c. Tremors, Panic Attacks, and Restlessness	36.4% (*8*/22)	10
10d. Numbness or Physical Insensibility	31.8% (*7*/22)	8
10e. Physical Exhaustion	9.1% (*2*/22)	2
10f. Acute Sleep and Eating Disorders	22.7% (*5*/22)	5
10g. Language Limitations (e.g., Mutism)	18.2% (*4*/22)	6
10h. Extreme Sensory Sensations (Hot/Cold)	9.1% (*2*/22)	2

## Data Availability

The data presented in this study are available on request from the corresponding author due to privacy reasons. Given the vulnerability of the adolescent population and the ethical responsibility to protect their privacy, complete anonymization of the transcripts would not be sufficient to ensure confidentiality. Therefore, the transcripts of the narrative interviews cannot be shared.

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
