# Peer review of "The Body as a Battleground: A Qualitative Study of the Impact of Violence, Body Shaming, and Self-Harm in Adolescents with a History of Suicide Attempts"

_ijerph, 2025, doi:10.3390/ijerph22060859_

Round 1

Reviewer 1 Report

Comments and Suggestions for Authors

The body as a battleground: a mixed method examination of the impact of violence, body shaming, and self-harm in adolescents with a history of suicide attempts

This study aims to explore how adolescents articulate body-related risk factors and warning signs, and contextualize them within their narratives of suicide attempts, and the importance that assign to their bodies and how they cope negative body-related experiences. 

I consider the topic is interesting and important. Congratulations on your research. I only have the next suggestions.

General suggestions

  • In the description of purpose, the authors do not address the quantitative variables they included in their descriptive analysis. I assume that most of them come from the qualitative analysis categories, but they also consider other variables from the patient records. It is important to mention them in this section.
  • I consider that your study does not conform to mixed methods research because you do not quantify and analyze enough variables that are reflected in the results. In my opinion, it is a qualitative study and I recommend you reclassify the design.
  • Perhaps some explanations of the categories can be summarized in the qualitative results.

Reviewer 2 Report

Comments and Suggestions for Authors

Thank you for the opportunity to read this paper. The research is quite interesting. In terms of the body experiences of trauma and the linkages to self-harm behaviours, this is very reminiscent of van der Kolk's work in his book The Body Keeps Score and Mate's work When the Body Says No.  Some linkage to that lineage of work would be appropriate as there is quite a bit of work around the ways in which trauma and related impacts link body and mental health. 

The linkages to internet are acutely interesting as it has a growing presence in this field - indeed the literature review might give that a bit more attention. It has more power now than other mediums.

As an academic and clinician, I found the work reinforces the profiles I have seen in a variety of settings. Thus, it is confirmatory and will help many students and early careers practitioners coming to make sense of this area of practice.

The reference to Axis 1 is likely referring to ICD but this should be explicit given that DSM 5 gave up the multiaxial approach.

The word cloud is an interesting approach to reporting findings - in my experience it is uncommon but again might well lead to interesting discussions in the classroom. I can well see using the paper in child and adolescent mental health courses. Indeed, I am likely to use it.

So, my recommended changes are few:

  1. Expand the literature related to trauma, body impacts, mental health and suicidality.
  2. Clarify ICD v DSM
  3. Expand the discussion on the impact of the internet.

Thank you for the work
